# Information Seeking for Robust Decision Making under Partial Observability

## Abstract

Explicit information seeking is essential to human problem-solving in practical environments characterized by incomplete information and noisy dynamics. When the true environmental state is not directly observable, humans seek information to update their internal dynamics and inform future decision-making. Although existing Large Language Model (LLM) planning agents have addressed observational uncertainty, they often overlook discrepancies between their internal dynamics and the actual environment. We introduce Information Seeking Decision Planner (InfoSeeker), an LLM decision-making framework that integrates task-oriented planning with information seeking to align internal dynamics and make optimal decisions under uncertainty in both agent observations and environmental dynamics. InfoSeeker prompts an LLM to actively gather information by planning actions to validate its understanding, detect environmental changes, or test hypotheses before generating or revising task-oriented plans. To evaluate InfoSeeker, we introduce a novel benchmark suite featuring partially observable environments with incomplete observations and uncertain dynamics. Experiments demonstrate that InfoSeeker achieves a $74\%$ absolute performance gain over prior methods without sacrificing sample efficiency. Moreover, InfoSeeker generalizes across LLMs and outperforms baselines on established benchmarks such as robotic manipulation and web navigation. These findings underscore the importance of tightly integrating planning and information seeking for robust behavior in partially observable environments.

## 1 Introduction

Real-world decision-making tasks are often partially observable, where observations and environmental dynamics may be noisy or uncertain. For example, in software engineering, a function may produce unexpected results due to incorrect usage or faulty implementation; In robotics, erroneous action control can result from wear and tear or inaccurate tuning of controllers. To correct failures, it is critical for agents to figure out the true underlying cause.

Humans exhibit strong problem-solving capabilities in dynamic and uncertain environments, enabled by two core abilities. Task-oriented planning selects action sequences to achieve a goal, while information seeking (Gopnik, 2012; Kidd & Hayden, 2015; Case & Given, 2016) proactively gathers information to align internal beliefs, inferred from internal dynamics, with the external world. Information seeking is especially crucial under partial observability, where the true environment state is hidden and decisions rely on imperfect beliefs. For example, to reach the target in Fig. 1a (blue block), we initially plan under the assumption that the robot arm follows the commanded (x, y) positions accurately. If the outcome deviates from expectations, we collect new evidence (e.g., moving to various locations and measuring the resulting positions) to update our beliefs about the environmental dynamics. Together, task-oriented planning and information seeking enable humans to uncover hidden causes and make robust decisions under uncertainty.

Large Language Models (LLMs) have emerged as versatile zero-shot agents for autonomous decision-making (Hu et al., 2023; Wang et al., 2023a; Hong et al., 2024). In interactive environments, they generate action sequences and revise strategies in response to feedback (Yao et al., 2023; Wang et al., 2023b). This closed-loop planning paradigm has shown promise across diverse domains such as robotics (Sun et al., 2023; Wang et al., 2024) and scientific discovery (Jansen et al., 2024). Recent studies have examined LLMs under partial observability, emphasizing either the recovery of hidden

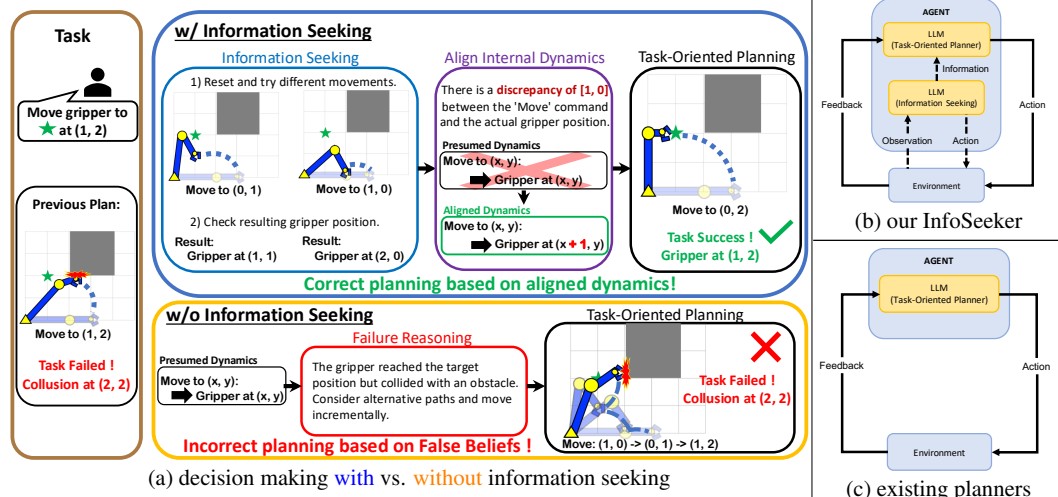

Figure 1: **Overcoming uncertainty through information seeking. (a)** When tasked to move robot gripper to a target location using miscalibrated controllers that introduce a constant (1, 0) offset to every command, existing planners (bottom) fail by over-relying on presumed dynamics without verifying them. In contrast, InfoSeeker (top) actively seeks information, detects discrepancies between commanded and executed motions, and updates its internal dynamics to generate a correct plan. **(b)** InfoSeeker validates its internal dynamics before planning, while **(c)** existing approaches depend solely on execution feedback and fixed assumptions. This difference enables InfoSeeker to succeed in environments with uncertain observations and dynamics.

knowledge (Ke et al., 2024; Krishnamurthy et al., 2024; Pan et al., 2025; Piriyakulkij et al., 2024) or identification of missing information in user instructions (Huang et al., 2024; Sun et al., 2024). However, these approaches overlook a critical challenge: mismatches between the agent's internal dynamics and the actual environment. Without informative observations to realign them, the agent develops inaccurate beliefs of latent states, leading to systematically flawed plans.

We propose Information Seeking Decision Planner (InfoSeeker), a framework that integrates information seeking directly into the decision-making loop. Our key insight is that robust decision making requires explicitly planned information seeking actions to reconcile the agent's internal dynamics with the external environment. As illustrated in Fig. 1a (blue block) and b, InfoSeeker prompts the LLM to actively gather information before proposing or revising a plan. Specifically, the LLM conducts targeted diagnostic trials that validate its understanding and detect shifts in environmental dynamics. In contrast, prior interactive planning approaches (Fig. 1a and c, yellow block) rely solely on reactive adaptation without explicit information gathering. By seeking evidence first, InfoSeeker uncovers the root causes of failure and adjusts its plans accordingly.

To evaluate our method's robustness, we introduce a suite of text-based simulation benchmarks that test LLM agents under partial observability and noisy dynamics. To our knowledge, this is the first benchmark that directly evaluates agents' planning capabilities under noisy environmental dynamics. Prior benchmarks (Fan et al., 2022; Shridhar et al., 2020; Wang et al., 2022) focus solely on observation uncertainty, where action outcomes are largely predictable. In contrast, our benchmarks incorporate uncertain dynamics, where actions may yield unexpected results due to unmodeled factors. For example, as illustrated in Fig. 1a, robot's final position can deviate from the commanded target due to miscalibrated controllers. This setting better reflects real-world scenarios, requiring agents to handle both incomplete observations and unpredictable dynamics that violate their assumptions. Although our benchmark is currently hand-crafted, it highlights the need for more rigorous evaluations of planning under uncertainty in both observations and dynamics.

InfoSeeker achieves an absolute performance gain of 74% over prior methods on our challenging benchmark. Active information seeking improves information acquisition without sacrificing sample efficiency, enabling the model to generate better task-oriented plans and achieve success faster. Moreover, InfoSeeker generalizes across different LLMs and outperforms existing approaches on

established benchmarks, including LLM3 (Wang et al., 2024) and TravelPlanner (Xie et al., 2024), demonstrating both versatility and robustness.

Our key contributions are: (1) An LLM-based planning framework that explicitly integrates information seeking to handle uncertainty in both dynamics and observations; (2) A novel benchmark suite for evaluating planning in partially observable environments with uncertainty in both observations and dynamics; (3) A formal connection between LLM-based planning and Partially Observable Markov Decision Processes (POMDPs).

## 2 PRELIMINARY

### 2.1 PARTIALLY OBSERVABLE MARKOV DECISION PROCESS

A Partially Observable Markov Decision Process (POMDP) models decision-making under uncertainty and is defined by the tuple $(\mathbb{S}, \mathbb{A}, \mathbb{Z}, T, O, R, \gamma)$, where $\mathbb{S}$ is a set of states, $\mathbb{A}$ is a set of actions, and $\mathbb{Z}$ is a set of observations. At each interaction step $t$, the environment is in a hidden state $s_t \in \mathbb{S}$. The agent takes an action $a_t \in \mathbb{A}$, transitions to a new state $s_{t+1} \sim T(\cdot \mid s_t, a_t)$, and receives an observation $o_{t+1} \sim O(\cdot \mid s_{t+1}, a_t) \in \mathbb{Z}$. Because the state is not directly observable, the agent infers the latent state by maintaining a belief $b_t \in \Delta(\mathbb{S})$—a distribution over states—updated via Bayes' rule from an initial belief state $b_0$:

$$b_t(s_t) = \frac{O(o_t \mid s_t, a_{t-1})}{P(o_t \mid b_{t-1}, a_{t-1})} \sum_{s_{t-1} \in \mathbb{S}} T(s_t \mid s_{t-1}, a_{t-1}) b_{t-1}(s_{t-1}) \tag{1}$$

with observation likelihood conditioned on previous belief $b_{t-1}$ and action $a_{t-1}$ as:

$$P(o_t \mid b_{t-1}, a_{t-1}) = \sum_{s_t \in \mathbb{S}} O(o_t \mid s_t, a_{t-1}) \sum_{s_{t-1} \in \mathbb{S}} T(s_t \mid s_{t-1}, a_{t-1}) b_{t-1}(s_{t-1}) \tag{2}$$

The sparse reward function $R : \mathbb{S} \times \mathbb{A} \to \{0, 1\}$ specifies the immediate reward, and the discount factor $\gamma \in [0, 1)$ determines the weighting of future rewards. The agent seeks to maximize expected return using the Bellman equation:

$$V^*(b_t) = \max_{a_t \in \mathbb{A}} \left[ \sum_{s_t \in \mathbb{S}} b_t(s_t) R(s_t, a_t) + \gamma \sum_{o_{t+1} \in \mathbb{Z}} P(o_{t+1} \mid b_t, a_t) V^*(b_{t+1}) \right] \tag{3}$$

where $P(o_{t+1} \mid b_t, a_t)$ is defined as in Eq. 2.

In our setting, the components $\mathbb{S}$, $\mathbb{Z}$, $T$, $O$, and $\gamma$ are unknown or only partially known. The agent is only given the action space $\mathbb{A}$ and the reward function $R$, both specified in natural language. We consider a zero-shot generalization scenario, where the agent perform on previously unseen tasks—each corresponding to a different POMDP—without any task-specific training. To obtain optimal belief state (Eq. 1) and make decisions that maximize return (Eq. 3), the agent must actively collect informative observations $\{o_1, o_2, \ldots, o_t\}$ and estimate transition probability $T(s_{t+1}|s_t, a_t)$.

### 2.2 LLM PLANNERS IN POMDP

Large Language Models (LLMs), pretrained on broad text corpora, encode structured knowledge that supports zero-shot reasoning and planning in novel environments. We interpret an LLM planner as a POMDP agent: given a task description and a trajectory $\tau_t = (o_0, a_0, o_1, \ldots, a_{t-1}, o_t)$, the LLM can reason about the latent states by constructing a belief $b_t$, and anticipate future changes in the environment via internal dynamics $T(s_{t+1}|s_t, a_t)$. Based on this, it generates a finite-horizon, task-oriented plan $a_{t:t+h-1}$ for maximizing expected return over $h$ steps.

While not explicitly trained for belief updating or value iteration, the LLM's reasoning and planning behavior can approximate Bayes' rule (Eq.1) and satisfy the Bellman equation (Eq.3). This allows zero-shot planning on novel tasks by leveraging prior knowledge and the input trajectory. However, if the observed trajectory $\tau_t$ lacks informative observations, or if the LLM's internal dynamics $T(s_{t+1}|s_t, a_t)$ deviates from the environment, the resulting belief state may be inaccurate and the expected return misestimate—leading to suboptimal plans.

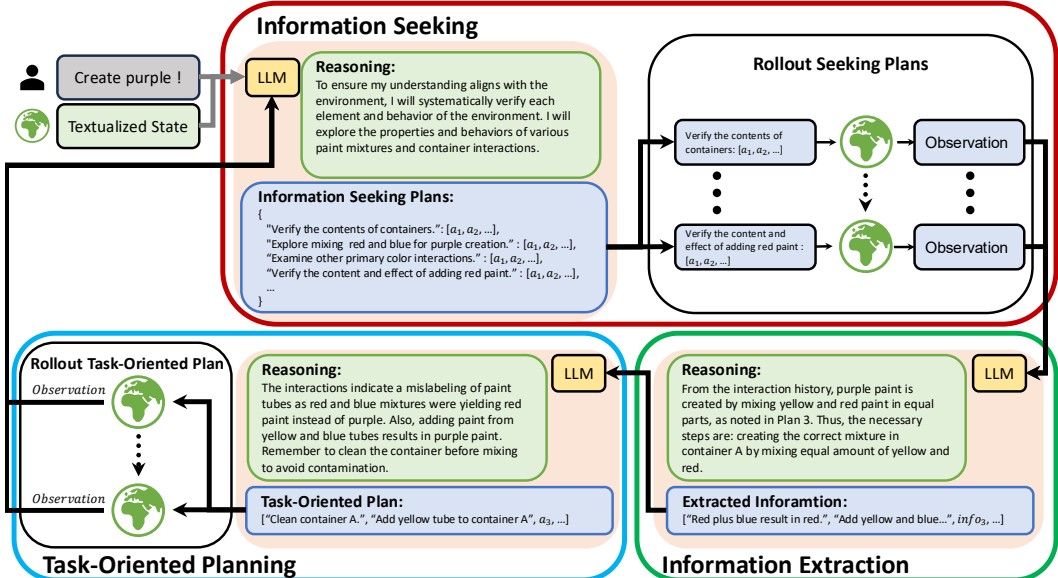

Figure 2: **System overview of InfoSeeker.** Our framework integrates *Information Seeking* (top-right) and *Task-Oriented Planning* (bottom-left) in a closed-loop process. The agent formulates and executes strategies to acquire missing knowledge, addressing gaps in its internal dynamics before generating more effective task plans. This iterative approach, supported by the reasoning capabilities of LLMs, enables the agent to reduce uncertainty and enhance planning effectiveness.

## 3 METHOD

We introduce InfoSeeker, an LLM agent that optimally estimates belief states by actively performing information seeking behaviors to collect informative observations and correct errors in the internal dynamics presumed by the LLM. As shown in Fig. 2, InfoSeeker performs iterative decision making: it first analyzes past trajectories to identify uncertainty and takes actions to gather information; next, it examines the resulting information seeking trajectories and refines task-oriented plans. By incorporating information seeking into the decision loop, InfoSeeker effectively refines its belief states and enhances future performance under uncertainty in both dynamics and observations.

### 3.1 INFORMATION SEEKING

Previous work (Huang et al., 2024; Sun et al., 2024) has focused primarily on uncovering missing information related to task instructions, often overlooking discrepancies between the agent's internal dynamics and the environment. In contrast, we propose a general prompting strategy that first guides the LLM to reason over current observations, prior plans, and interaction history, and then to identify targeted exploratory actions. When inconsistencies are detected, the agent can hypothesize potential errors, design focused experiments to test its assumptions, or detect shifts in environmental dynamics. This reasoning-driven process supports both the verification of the internal dynamics and the acquisition of informative observations, allowing for more accurate belief state update and more effective task-oriented planning.

After executing exploratory actions, the agent receives a new trajectory which it uses to update internal dynamics and refine belief states for subsequent planning. In practice, we introduce an information extraction module that prompts an LLM to analyze the information seeking trajectory, extract key insights, and generate concise summaries. These summaries are then incorporated into the context of subsequent task-oriented planning. We refer to Appendix G.1 for details of our prompt.

## 3.2 Task-Oriented Planning

Based on the extracted insights, the LLM is prompted to generate an initial plan or revise a previous one to complete the task. Following Wang et al. (2024), our prompting strategy guides the LLM to reason over the combined context before generating a new sequence of task-oriented actions. Once these actions are executed, the updated trajectory is passed back to the LLM, initiating the next cycle of information seeking and task-oriented planning.

This iterative process continues until the task is completed or a predefined number of interaction steps is reached. By continually collecting informative observations and refining its internal dynamics, InfoSeeker adapts effectively to partially observable environments with uncertain or shifting dynamics. Full methodological details are provided in Algorithm 1 (Appendix B).

## 4 Benchmarking Robustness in POMDPs with Noisy Dynamics

We introduce a suite of text-based simulation benchmarks for evaluating the robustness of LLM agents in partially observable environments with noisy environmental dynamics. Existing benchmarks (Fan et al., 2022; Shridhar et al., 2020; Wang et al., 2022) consider only uncertainty in observations, where actions always yield predictable outcomes (e.g., "turn on lamp" action exactly turns on the lamp). In contrast, we consider incorporating uncertain dynamics: actions may fail due to unmodeled factors, such as control errors or environmental noise. This setup highlights the critical challenges of decision-making in real-world scenarios. To contrast agent performance under both certain and uncertain dynamics, we implement each task in two configurations: a *Basic* version, which resembles existing benchmarks, and a *Perturbed* version that additionally incorporates noisy dynamics.

### 4.1 Task Suite Overview

Our benchmark contains a suite of 5 tasks, spanning a wide range of decision making domains. We detail the configuration of each task as following:

**Robot arm control.** A 2D motion planning task in which the agent controls a planar 2-joint robot arm to move its end-effector to a target location while avoiding static obstacles. In the *Basic* condition, actions are executed as intended. In the *Perturbed* condition, a constant offset $(\Delta x, \Delta y)$ is added to each commanded action, simulating real-world uncertainty in system dynamics due to actuation bias or calibration errors. Agents must infer this transition discrepancy and adjust its planning accordingly.

**Robot navigation.** A long-horizon task requires a mobile robot to reach a ball, pick it up, and navigate to a designated goal location. The agent controls the robot using four directional actions: forward, backward, left, and right. In the *Basic* condition, each action produces the expected movement. In the *Perturbed* condition, uncertain dynamics is introduced by inverting the action mappings—for example, the "left" action causes the robot to move right. To complete the task, the agent must detect and adapt to these inconsistencies between its commands and the robot's actual behavior.

**Mix colors.** In this task, the agent must mix paints from a set of labeled tubes in containers to produce a target color, following a pigment-based mixing rule (Sochorová & Jamriška, 2021). In the *Basic* condition, tube labels accurately reflect their contents, and the container is initially clean. In the *Perturbed* setting, we introduce two sources of uncertainty: (1) the container may be pre-contaminated with unknown colors, and (2) tube labels may be incorrect. The agent must infer the true state of the environment from noisy observations and reason about uncommon action outcomes.

**Block stacking.** This task is inspired by classic BlocksWorld scenarios (Valmeekam et al., 2023), where the agent must rearrange stacks of colored blocks to match a target configuration. The agent can only pick up the top block from a stack or from a limited-capacity inventory (holding at most one block), and may only place blocks on top of a stack or into the inventory. We define two difficulty levels: a simple version with a single target stack, and a complex version requiring coordination across multiple stacks and longer planning horizons. In the *basic* condition, the agent has complete and accurate knowledge of the initial inventory. In the *perturbed* condition, the inventory state is initially unknown, requiring the agent to inspect it before executing a plan. This setting tests the agent's ability to reason under uncertain observation while performing precise, goal-directed actions.

| | robot arm control | | mix color | | | robot navigation | | stack multiple blocks | | stack single block | |
|---|---|---|---|---|---|---|---|---|---|---|---|
| | basic | perturbed | basic | contaminated | wronglabel | basic | perturbed | basic | perturbed | basic | perturbed |
| *Gemini Flash 2.0* | | | | | | | | | | | |
| ReAct | 48 | 2 | 48 | 32 | 2 | **100** | 18 | 30 | 6 | 68 | 22 |
| AdaPlanner | 12 | 0 | 40 | 16 | 2 | 76 | 0 | 14 | 2 | 14 | 0 |
| LLM3 (backtrack) | **100** | 6 | 64 | 60 | 6 | 96 | 4 | **50** | 26 | 76 | 6 |
| LLM3 (from scratch) | 98 | 2 | 54 | 44 | 4 | **100** | 4 | 42 | **36** | **86** | 0 |
| InfoSeeker (ours) | **100** | **80** | **76** | 80 | 14 | **100** | 46 | 42 | 34 | 82 | **62** |
| *GPT 4o* | | | | | | | | | | | |
| ReAct | 22 | 2 | 42 | 3 4 | 6 | 90 | 4 | 18 | 10 | 54 | 20 |
| AdaPlanner | 80 | 2 | 38 | 24 | 2 | 82 | 0 | 16 | 12 | 22 | 10 |
| LLM3 (backtrack) | **100** | 12 | 74 | 76 | 10 | 80 | 6 | 38 | 30 | 80 | 4 |
| LLM3 (from scratch) | **100** | 6 | 76 | 72 | 6 | **96** | 8 | **48** | 50 | 56 | 4 |
| InfoSeeker (ours) | **100** | **22** | **84** | 78 | **36** | 92 | **44** | 38 | 26 | **84** | 34 |

Table 1: **Quantitative results on the proposed benchmark.** We report the success rate (%), defined as the proportion of tasks successfully completed within 100 interaction steps per task instance. We evaluate two LLMs across five planning methods. By incorporating active information seeking, InfoSeeker consistently achieves higher success rates, particularly under perturbed conditions.

## 5 EMPIRICAL RESULTS

This section details the empirical evaluation of the proposed InfoSeeker in partially observable environments. Our primary objectives are to: (1) assess its efficacy and generalization under uncertainty in observations (Sec.5.2); (2) quantify its performance under uncertainty in both observations and dynamics (Sec.5.3); (3) evaluate the efficiency of iterative planning (Sec.5.4); and (4) distinguish the contributions of each method component through ablation studies (Sec.5.5). We benchmark InfoSeeker against prior LLM-based planning methods using both our proposed tasks and two established benchmarks: LLM3(Wang et al., 2024) and TravelPlanner(Xie et al., 2024).

### 5.1 EXPERIMENTAL SETUP

Experiments were conducted across 11 interactive partial observable tasks proposed in our benchmark. These tasks are designed to evaluate iterative planning capabilities under POMDP settings. We adopt success rate–the percentage of tasks successfully completed within 100 interaction steps per task–as the evaluation metric, where interaction steps refer to action taken to interact with the environment.

We compared InfoSeeker against several established LLM planning methods: **ReAct** (Yao et al., 2023), a widely adopted baseline that interleaves reasoning traces and task-specific action generation. **AdaPlanner** (Sun et al., 2023), a code-style planner that adaptively refines self-generated plans based on environmental feedback. **LLM3** (Wang et al., 2024), a text-based planner that refines strategies based on previous planning traces. Our implementation uses the last 5 traces, consistent with the original publication. We implemented two variants: *backtrack*, which reverts to the last successful action before generating a new plan, and *from scratch*, which discards all previous plans and generates a new one.

To evaluate robustness and generalization, all methods were tested across two LLMs: Gemini Flash 2.0 (AI, 2025) and GPT-4o (Hurst et al., 2024). We refer to Appendix G for further implementation details and prompt examples.

### 5.2 PERFORMANCE UNDER UNCERTAINTY IN OBSERVATIONS

We evaluated InfoSeeker under uncertainty in observations. We test on the *basic* setting of our proposed benchmark, as well as two established benchmarks: **LLM3** and **TravelPlanner**.

As shown in in Table 1, our InfoSeeker demonstrates competitive performance against strong baselines in *basic* setup of our benchmark. It is not surprising that all baselines perform well in the *basic* setup, as they have already been evaluated on more complex and larger-scale benchmarks. In contrast, our method is general. It achieves consistent yet marginal performance gain across most tasks over

these baselines. In the *mix color* task, InfoSeeker reached an $84\%$ success rate, obtaining an absolute performance gain of $8\%$ over the best baseline.

| | Setting 1 | | | Setting 2 | | |
|---|---|---|---|---|---|---|
| | Easy | Medium | Hard | Small | Medium | Large |
| LLM3 (backtrack) | 100 | 80 | 50 | 50 | 80 | 50 |
| LLM3 (from scratch) | 100 | 100 | 80 | 80 | 90 | 60 |
| InfoSeeker (ours) | 100 | 100 | 70 | 70 | **100** | **80** |

Table 2: **LLM3 benchmark (Wang et al., 2024).** The benchmark requires agents to control a robotic arm to pack objects into a basket in a simulated environment. In Setting 1, all areas of the basket are fully reachable. In Setting 2, objects are fixed at the hardest level, and a larger basket size increases unreachable regions

| | Final Pass Rate ($\uparrow$) |
|---|---|
| ReAct (using original prompt) | 5.56 |
| LLM3 (backtrack) | 5.00 |
| LLM3 (from scratch) | 5.56 |
| InfoSeeker (ours) | **6.11** |

Table 3: **TravelPlanner benchmark (Xie et al., 2024).** The benchmark evaluates web agents on their ability to gather user-specific information. All methods are evaluated with a fixed 100 interaction steps.

We additionally evaluate InfoSeeker on two established benchmarks. First, we test on a robotic manipulation benchmark introduced by LLM3, where agents control a robotic arm in a physical simulation. This task emphasizes challenges in spatial reasoning. To ensure a fair comparison, we follow the original evaluation protocol, allowing up to 20 planning attempts per task. As shown in Table 2, InfoSeeker consistently outperforms LLM3 baselines across all setups. Notably, in the most difficult scenario (*Large* basket size in Setting 2) InfoSeeker achieves a $20\%$ absolute performance gain. Next, we test on TravelPlanner benchmark, where web agents gather information based on user preferences to propose travel plans. As shown in Table 3, InfoSeeker surpasses all baselines in final pass rate. In particular, our method can identify the ambiguity of user instruction. For example, it tried to clarify whether Washington refers to Washington state or Washington, D.C. while other methods fail to identify this ambiguity. These results demonstrated InfoSeeker's strong generalization capabilities in more realistic and challenging use cases.

| | basic | perturbed |
|---|---|---|
| *With Uncertain Prompt* | | |
| ReAct | 62 | 28 |
| AdaPlanner | 8 | 4 |
| LLM3 (backtrack) | 80 | 12 |
| LLM3 (from scratch) | **90** | 12 |
| InfoSeeker (ours) | 82 | **62** |

| | basic | perturbed |
|---|---|---|
| ReAct | 68 | 30 |
| AdaPlanner | 10 | 0 |
| LLM3 (backtrack) | 98 | 46 |
| LLM3 (from scratch) | 98 | 62 |
| InfoSeeker (ours) | **100** | **98** |

Table 4: **Explicit uncertainty in prompts yields minimal gains.** On the *stack single block* task, adding uncertainty descriptions to baseline prompts results in only marginal improvement. This highlights the limitations of prompting alone and the necessity of information-seeking.

Table 5: **Time efficiency of InfoSeeker.** We evaluate all methods on the *stack single block* task, with a fixed wall-clock time of 1 minute per trial. InfoSeeker demonstrates the highest efficiency, achieving a $36\%$ performance improvement over the previous state-of-the-art, LLM3.

### 5.3 PERFORMANCE UNDER UNCERTAINTY IN OBSERVATIONS AND DYNAMICS

InfoSeeker demonstrates strong performance in environments with uncertain environmental dynamics that may be inconsistent with the agents' assumptions. This setting requires adaptation of the agent's internal dynamics to the current environment. We test on the *perturbed* settings of our benchmark, and as shown in Table 1, InfoSeeker consistently outperforms all baselines. Take *robot arm control* task as an example, where the robotic arm controller is miscalibrated, InfoSeeker achieved an $80\%$ success rate. In contrast, the performance of the best-performing baseline in *basic* setting dropped from $100\%$ to $6\%$. The substantial performance gain attributes to our explicit integration of information-seeking behaviors, rather than prompt engineering. As shown in Table 4, we provide baselines with the same description of environmental uncertainty as InfoSeeker. These baselines do not benefit from such prompts, obtaining only marginal performance gain in *stack single block* task. These results underscore the importance of explicit information seeking behavior, allowing InfoSeeker to validate

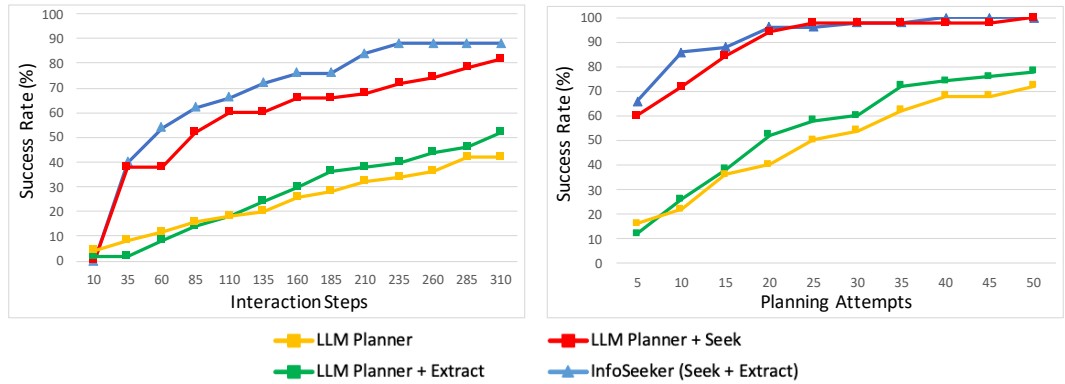

Figure 3: **Ablation study.** Success rate (%) versus interaction steps (left) and planning attempts (right) on the *perturbed stack single block* task. Combining information-seeking and information-extraction behaviors makes our InfoSeeker more efficient and effective.

internal dynamics, refine belief states, and avoid repeating flawed reasoning patterns. We present visual examples of planned trajectories in Appendix C.

## 5.4 MODEL EFFICIENCY

As InfoSeeker actively gather information to correct belief states, it is more time- and resource-efficient than prior arts in POMDP settings. Although information seeking behaviors require additional interaction with the environment, it allows InfoSeeker to better diagnose the causes of failure and generate optimal plans. As shown in Table 5, InfoSeeker surpass all baselines under a fixed one-minute wall-clock execution time in *stack single block* task. It outperforms LLM3 baselines by 36% in the *perturbed* setting. Our additional experiments, that calculate task success rates across varying numbers of interaction steps and planning attempts, further demonstrate the efficiency of our InfoSeeker. We refer to Appendix D for more details.

## 5.5 ABLATION STUDIES

We conduct ablation studies to assess the contributions of each component of InfoSeeker using the *perturbed stack single block* task. We evaluate the model's performance according to the number of interaction steps. As shown in the left panel of Figure 3, the vanilla LLM planner achieves a 42% success rate after 310 interaction steps. Incorporating information extraction to analyze trajectories from previous task-oriented planning (*Extract*) yields only marginal improvement. In contrast, prompting the LLM to engage in explicit information seeking behavior (*Seek*) leads to a substantial performance boost, increasing the success rate to 82%. InfoSeeker, which combines both information seeking and information extraction from the exploratory trajectory, achieves the highest success rate. It reaches 72% using just 135 interaction steps—nearly halving the steps needed by *Seek* to reach similar performance.

Meanwhile, we also evaluate the iterative planning method according to the number of planning attempts. As shown in the right figure of Figure 3, the combination of information seeking and information extraction behaviors is key to our InfoSeeker.

Lastly, we verify if information seeking behaviors emerge from in-context learning. As shown in Appendix E, we provide the vanilla LLM planner with successful demonstrations of our InfoSeeker on *mix color* task. However, these context prompts only bring marginal performance gain. Our results necessitate explicit integration information-seeking behaviors in the decision-making loop.

## 6 RELATED WORK

### 6.1 LLM PLANNING AGENTS

Foundational works like Inner Monologue (Huang et al., 2022) and ReAct (Yao et al., 2023) demonstrated that LLMs could process execution feedback and scene descriptions to revise plans. Subsequent methods, including ProgPrompt (Singh et al., 2023) and AdaPlanner (Sun et al., 2023), sought to reduce plan ambiguity by generating plans in programming languages. These approaches have been applied to interactive, partially observable domains, including open-world embodied agents (Wang et al., 2023a;b; Zhu et al., 2023) and real-world robotic systems (Wang et al., 2024; Ding et al., 2023; Chen et al., 2024; Joublin et al., 2024). Beyond planning, several studies investigate how LLMs gather and reason about information, using benchmarks inspired by human cognitive tasks (Ke et al., 2024; Krishnamurthy et al., 2024; Pan et al., 2025; Piriyakulkij et al., 2024). Other research focuses on task-specific exploration, such as identifying missing information in instructions (Huang et al., 2024; Sun et al., 2024), or seeking human assistance (Li et al., 2023; Chen et al., 2023). Memory-augmented agents (Shinn et al., 2023; Zhao et al., 2024; Sarch et al., 2023; Song et al., 2024; Sarch et al., 2024) enhance performance by replaying and learning from past trajectories. These methods typically address uncertainty in observability while assuming known or static environmental dynamics. This reduces their effectiveness in settings with uncertainty in both observation and environmental dynamics, where they lack mechanisms to validate prior knowledge and adapt internal dynamics. In contrast, InfoSeeker actively seeks out informative interactions to reduce uncertainty, enabling more efficient adaptation to incomplete observations and shifting dynamics.

### 6.2 EVALUATION IN INTERACTIVE ENVIRONMENTS.

A variety of benchmarks have been developed to evaluate the planning and reasoning capabilities of LLMs. Valmeekam et al. (2023) focus on structured reasoning tasks, while text-based environments (Shridhar et al., 2020; Côté et al., 2019; Chevalier-Boisvert et al., 2023) assess decision-making under incomplete observation. Long-horizon and open-ended reasoning are evaluated in complex simulations such as game environments (Fan et al., 2022; Küttler et al., 2020), scientific discoveries (Wang et al., 2022; Jansen et al., 2024), and web navigation (Yao et al., 2022; Deng et al., 2023; Xie et al., 2024). Although many of these benchmarks involve partial observability, they typically focus solely on uncertainty in observations. In contrast, our benchmark captures more realistic conditions: observations are incomplete, and environmental dynamics are noisy. This setting requires agents to adapt their decision-making under limited information and uncertain dynamics.

## 7 LIMITATION AND CONCLUSION

**Limitations.** We acknowledge that the proposed benchmark is relatively small-scale and hand-crafted. However, to the best of our knowledge, there is no existing benchmark that evaluates the robustness of models under uncertain environmental dynamics. We believe that creating a rigorous, large-scale benchmark is an important direction for future work. Additionally, the information extraction component in InfoSeeker can occasionally extract misleading or irrelevant insights, which may negatively impact the planner (see Appendix F for details). Enhancing the accuracy and reliability of information extraction is a key direction for improvement.

**Conclusion.** We propose a novel decision-making framework for LLM agents in partially observable environments with noisy dynamics. Our method integrates explicit information seeking and task-oriented planning within each decision cycle, allowing agents to adapt their internal dynamics and update belief states more accurately. To evaluate robustness, we introduce a text-based simulation benchmark that tests LLM agents under limited observability and uncertain environmental dynamics. Experiments show that InfoSeeker outperforms baselines in adapting internal dynamics and generating robust plans. Additional evaluations on established benchmarks demonstrate strong generalization. By embedding explicit information seeking into the decision loop, InfoSeeker improves planning under uncertainty. See Appendix A for our disclosure of LLM usage.

## REPRODUCIBILITY STATEMENT

The complete implementation of our proposed InfoSeeker along with the benchmarks is included in the supplementary material. While the implementations of baseline methods and ablation experiments are not included, we provide the corresponding prompts used with LLMs in Appendix G.

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

APPENDIX

## A    DECLARATION OF LLM USAGE

We used large language models (LLMs) to assist with writing refinement (e.g., grammar, spelling, word choice) and to support the design and execution of experiments. Our paper proposes a novel LLM-based framework for autonomous decision planning, making LLM a central component of our methodology; this framework is detailed in Section 3 and Algorithm 1. The specific LLMs used are listed in Section 5.1.

## B    INFOSEEKER ALGORITHM

---

**Algorithm 1:** Information Seeking Decision Planner (InfoSeeker)

---

**Input:** a LLM $f$, task instruction $l$, prompt for information seeking $p_s$, information extraction $p_e$, and task-oriented planning $p_t$, maximum number of planning attempts $N_{max}$, maximum number of interaction steps $K_{max}$, environment $e$

**Output:** a sequence of task-oriented actions $A_t$

Initiate interaction history $H \leftarrow \{\}$ and the total number of interaction steps $K \leftarrow 0$

**for** *planning attempt* $n \in (1, ..., N_{max})$ **do**

    Generate actions for information seeking $A_s = f(p_s, l, H)$

    Execute actions and append new observations to history $H \leftarrow H \cup \{(a, e(a)) | a \in A_s\}$

    Update the total number of interaction steps $K \leftarrow K + |A_s|$

    Extract information $I = f(p_e, l, H)$

    Generate task-oriented plans $A_t = f(p_t, l, H, I)$

    Reset interaction history $H \leftarrow \{\}$

    Execute actions and append feedback to history $H \leftarrow H \cup \{(a, e(a)) | a \in A_t\}$

    Update the total number of interaction steps $K \leftarrow K + |A_t|$

    **if** *success **or** $K \geq K_{max}$* **then**

        |   break;

    **end**

**end**

**return** $A_t$

---

## C    PLANNING VISUALIZATION

The adaptive planning process is illustrated in Figure 4, comparing the behaviors of InfoSeeker and LLM3 when planning *from scratch*. While LLM3 repeatedly applies an invalid reasoning pattern—such as attempting to move "forward then right," even though this consistently results in the robot moving "backward then left" (Figure 4c, bottom row)—InfoSeeker systematically explores alternative actions and adjusts its strategy accordingly. For example, it first probes all four directions to map actions to their actual outcomes, then uses this knowledge to adapt effectively on subsequent attempts (see Figure 4c, top row). This process demonstrates InfoSeeker's ability to actively acquire informative observations and adapt its internal dynamics to align with the environment.

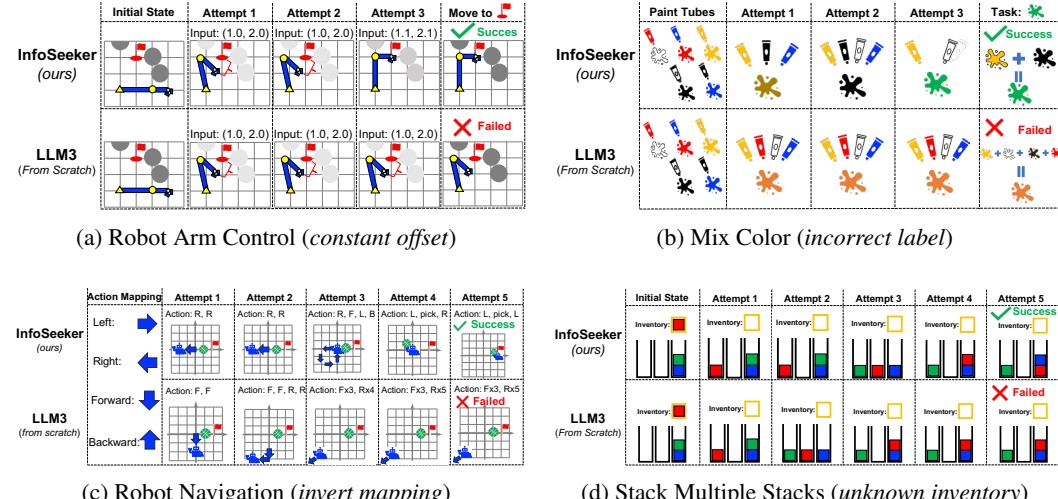

(a) Robot Arm Control (*constant offset*)  (b) Mix Color (*incorrect label*)

(c) Robot Navigation (*invert mapping*)  (d) Stack Multiple Stacks (*unknown inventory*)

Figure 4: **Visualization of InfoSeeker and LLM3 (*from scratch*) in perturbed environments. (a)** Robotic arm guidance to target $(1.0, 2.0)$ with a fixed action offset $(-0.1, -0.1)$. **(b)** Paint mixing for seagreen (yellow + black) using mislabeled tubes: the "red" tube contains white, "blue" contains red, "white" contains black, and "black" contains blue. **(c)** Navigate to ball at $(1, 0)$ and deliver it to goal location $(2, 0)$ under inverted action mappings ("left" moves right, "forward" moves backward). **(d)** Rearranging blocks to match a target configuration, starting with a hidden inventory (contains a red block). Across these perturbations, InfoSeeker adapts through information seeking and feedback, while LLM3 repeatedly failed due to persistent misinterpretation.

## D  PERFORMANCE WITH VARYING INTERACTION STEPS AND PLANNING ATTEMPTS

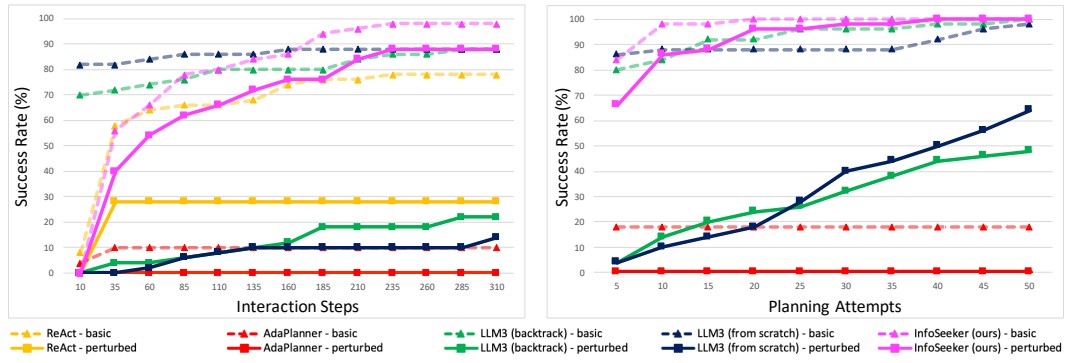

Figure 5: **Performance with varying interaction steps (*left*) and planning attempts (*right*).** The plots show the success rate (%) of InfoSeeker and four LLM baselines Yao et al. (2023); Sun et al. (2023); Wang et al. (2024) on the stack single block task, as interaction steps (*left*) and planning attempts (*right*) are varied. ReAct is included only in the interaction step analysis due to its lack of distinct planning attempts. The plots demonstrate that InfoSeeker effectively leverages increased resources, particularly in perturbed environments.

We evaluated InfoSeeker's performance in the *stack single block* task by measuring its success rate under varying interaction steps and planning attempts (Figure 5). The results demonstrate that InfoSeeker effectively leverages increased resources, particularly in perturbed environments. Figure 5(left) shows the impact of interaction steps; increasing interaction steps in the perturbed setting leads to a substantial performance gain—from $0\%$ at 10 steps to $88\%$ at 235 steps. Notably, this success rate matches the best-performing baseline (LLM3 *from scratch*) in the much easier

*basic setting*. A similar trend appears when varying planning attempts (Figure 5, right). InfoSeeker achieves a 66% success rate with just 5 attempts in the perturbed task, significantly outperforms leading baselines LLM3 *from scratch* and *backtrack*, which reach just 4%. As planning attempts increase, InfoSeeker continues to improve, reaching 100% success after 40 attempts. These findings highlight InfoSeeker's strong adaptability through active information seeking and the adaptation of its internal dynamics, enabling robust performance under uncertainty in both observations and environmental dynamics.

## E   VANILLA LLM PLANNER WITH IN-CONTEXT LEARNING

We carried out experiments on *mix color* task, where a vanilla LLM planner was guided using in-context demonstrations of successful InfoSeeker execution traces in perturbed environments. As shown in the Table 6, the in-context examples only brings marginal performance gain to the planner. They do not induce effective information seeking behaviors. In particular, we observed that the planner fails to follow the provided examples and cannot align its internal dynamics with the environment, as evidenced by its inability to detect incorrect paint labels. Our results demonstrate that abstracting information seeking behaviors from in-context examples is challenging.

|                     | Basic | Contaminated | Wrong Label |
|---------------------|-------|--------------|-------------|
| Vanilla Planner     | 52    | 46           | 6           |
| In-Context Learning | 58    | 56           | 2           |
| InfoSeeker (ours)   | **76**| **80**       | **14**      |

Table 6: **In-context learning fails to induce information-seeking behavior.** We evaluate the *mix color* task, where a planner is guided by successful InfoSeeker demonstrations in perturbed environments. In-context examples yield only marginal gains, indicating that information-seeking cannot be effectively learned through in-context learning.

## F   FAILURE ANALYSIS

To understand the limitations of InfoSeeker, we present an analysis of failure cases in *block stacking* tasks. We categorize failure cases into four types: (1) **Information Seeking**, where the agent fails to acquire informative observations needed to detect the mismatches between its internal dynamics and the environment; (2) **Information Extraction**, where agent extracts misleading or irrelevant information that misguide the planner; (3) **Instruction Understanding**, where the agent misinterprets user instructions (e.g., confusing "red on top of blue" with "blue on top of red"); and (4) **Long-horizon Planning**, where the agent detects a mismatch between internal and environmental dynamics during the information seeking phase, but fails to produce effective actions to complete the task. As shown in the Figure 6, extracting incorrect information is the major failure type of InfoSeeker, especially in *stack multiple blocks task*. How to enhance the quality of information extraction will be an interesting future work.

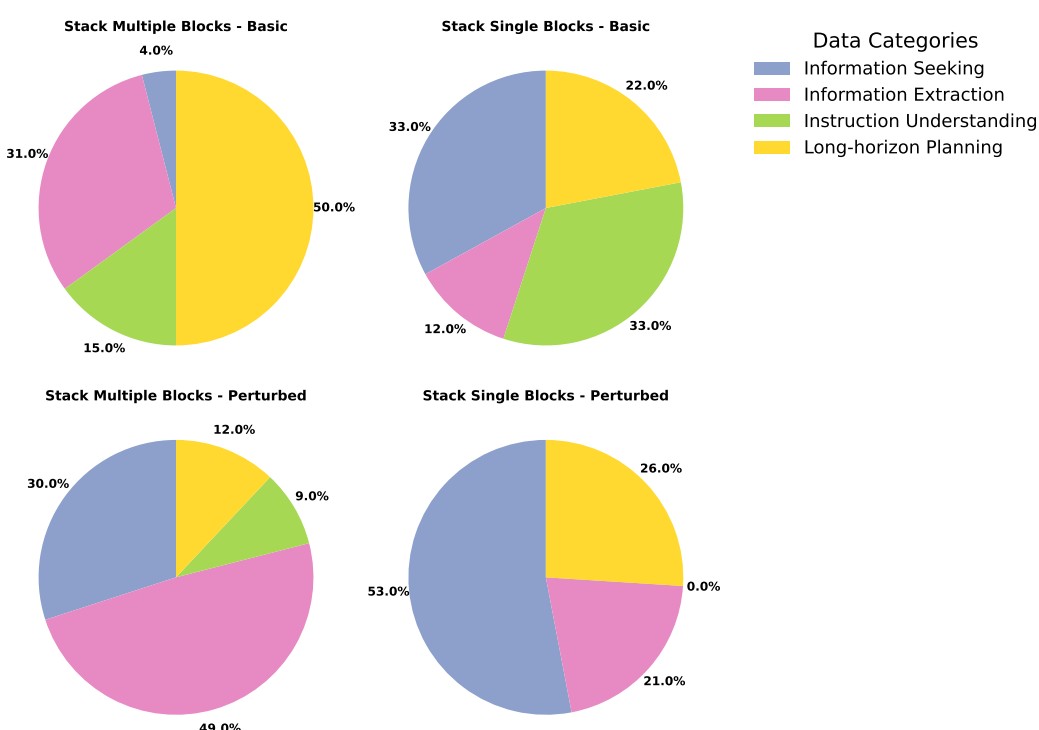

Figure 6: **Failure Analysis.** Extracting incorrect information is the main failure of InfoSeeker, particularly in the *stack multiple blocks task*. Improving information extraction quality is a promising direction for future work.

## G  PROMPTS AND IMPLEMENTATION DETAILS

ReAct and AdaPlanner were evaluated using few-shot prompting, in line with their original publications. While both LLM3 variants and InfoSeeker used zero-shot prompting. We used API-default hyperparameters (e.g., temperature, top_p) for all experiments to ensure comparability among methods, though performance may improve with dedicated hyperparameter tuning. The prompts are present in InfoSeeker G.1, ReAct G.2, AdaPlanner G.3, LLM3 G.4, and In-Context Learning G.5. To add description of environmental uncertainty to baseline prompts we use the same words as InfoSeeker in Listing 1.

```
You are an AI robot that generate a plan of actions to reach the goal.
You will be given a domain description and the trace of generated plans
    that lead to motion failure.
Previous actions have not completed the goal, possibly due to
    misunderstandings of the environment, insufficient exploration, or
    flaws in your earlier plan.
```

Listing 1: Prompt for Uncertainty

## G.1 INFOSEEKER PROMPTS

```
You are an AI tasked with verifying your understanding of a given
    environment and exploring it for additional information.
Based on the provided domain description, generate a plan to ensure your
    understanding is aligned with the real environment and how you will
    explore it for further details.
Try to verifying your understanding thoroughly, and explore beyond the
    goal for unexpected solution.
You are expected to learn from the environment only, do not try to
    achieve the goal.

Important: You have VERY FEW turns left. Choose your next action
    carefully to maximize information.

# Domain Description
{domain_desc}

Please provide the following output:
1. **Reasoning**: Explain the strategy you will use to verify your
    understanding of the environment and explore it for new information.
2. **Steps**: Provide a detailed series of steps you will take to verify
    and explore the environment. Make sure to reset and clean the
    environment at the end of each step to avoid any potential
    interference. Each step should include:
    - **Goal**: The specific objective or target for that step (what you
        aim to achieve).
    - **Action Plan**: A list of actions that you will perform to
        accomplish the goal of that step.

Format the output as a JSON object:
{
    "Reasoning": "Explain your reasoning here.",
    "Steps": [
        {
            "Goal": "The goal of your verification or exploration",
            "Action Plan": ["action1", "action2", "action3", ...]
        },
        ...
    ]
}
```

Listing 2: Information-seeking prompt for InfoSeeker in our benchmark when no prior plan exists

```
You are an AI tasked with generating a plan to achieve a given goal.
You will be given a domain description and a history of past actions that
    have not yet achieved the goal.
You must analyze the interaction history, understand the environment
    mechanics, and determine why the goal has not been reached.
Using this insight, create a new plan to successfully achieve the goal.

# Domain Description
{domain_desc}

# Interaction History
{interaction_history}

# Information
{information}

Please provide the following output:
```

```
1. **Reasoning**: Describe the strategy you will use to achieve the goal,
    taking into account the past actions and information in the
   interaction history. Consider how the environment's response to these
    actions affects your new plan.
2. **Solution Plan**: List the series of actions you will take to achieve
    the goal, ensuring that your actions align with the reasoning you
   provided. Remember to check for the final result.

Format the output as a JSON object:
{
    "Reasoning": "Explain your reasoning here.",
    "Solution Plan": ["action1", "action2", "action3", ...]
}
```

Listing 3: Task-Oriented Planning prompt for InfoSeeker in our benchmark

```
You are an AI tasked with revising your plan to achieve a goal.
Previous actions have not completed the goal, possibly due to
    misunderstandings of the environment, insufficient exploration, or
    flaws in your earlier plan.
You can conduct simple tests to identify issues with the past approach,
    verify your understanding, and explore the environment for more
    information if needed.
Using the provided domain description and interaction history, generate
    an action plan to verify and explore the environment, identifying
    flaws in previous actions.
Try to verify the failures by simple unit tests, or explore beyond past
    plans for unexpected solution.

Important: You have VERY FEW turns left. Choose your next action
    carefully to maximize information.

# Domain Description
{domain_desc}

# Interaction History
{interaction_history}

Please provide the following output:
1. **Reasoning**: Explain your strategy for verifying your understanding
    of the environment, exploring it for new insights, and addressing
    issues in the previous plan.
2. **Steps**: Provide a detailed series of steps you will take to verify
    and explore the environment. Make sure to reset and clean the
    environment at the end of each step to avoid any potential
    interference. Each step should include:
   - **Goal**: The specific objective or target for that step (what you
       aim to achieve).
   - **Action Plan**: A list of actions that you will perform to
       accomplish the goal of that step.

Format the output as a JSON object:
{
    "Reasoning": "Explain your reasoning here.",
    "Steps": [
        {
            "Goal": "The goal of your verification or exploration",
            "Action Plan": ["action1", "action2", "action3", ...]
        },
        ...
    ]
}
```

Listing 4: Information Seeking prompt for InfoSeeker in our benchmark when prior plan exists

## G.2    REACT PROMPT

```
Interact with materials to solve a task. Here are two examples.
# Example 1
Your task is to: move the gripper to [1.0, 0.0], while avoid collision
    with the obstacles at, [2.0, -2.0] radius 0.5
Initial State:
Joint positions: 'Joint 0': [0.00, 0.00] 'Joint 1': [2.00, 0.00] 'Gripper
    ': [3.00, 0.00]
> think: To solve the task, I need to first check the action options.
OK.
> help
1) Move x y: Move the gripper to (x, y). The rotation of the joints will
    be calculated by inverse kinematics
2) Check: Check Robot arm joint positions
3) Help: View the available action options.
4) think: <your_thought>: Output your thinking and reasoning.
5) End: End the task when you finished, or the task is not solvable.
> think: To solve the task, I need to move gripper to [1.0, 0.0].
OK.
> Move 1.0 0.0
Success!
> Check
Joint positions: 'Joint 0': [0.00, 0.00] 'Joint 1': [2.00, 0.00] 'Gripper
    ': [1.00, 0.00]
> think: Now the gripper is at [1.0, 0.0], the task is complete.
OK.
> End
# Now it's your turn
Your task is to: move the gripper to [1.0, 0.0], while avoid collision
    with the obstacles at, [0.0, 3.0] radius 0.5, [2.0, 2.0] radius 0.5,
    [-2.0, 2.0] radius 0.5, [2.5, 1.0] radius 0.5, [-2.5, 1.0] radius
    0.5, [0.0, -3.0] radius 0.5, [2.0, -2.0] radius 0.5, [-2.0, -2.0]
    radius 0.5, [2.5, -1.0] radius 0.5, [-2.5, -1.0] radius 0.5
Initial State:
Joint positions: 'Joint 0': [0.00, 0.00] 'Joint 1': [2.00, 0.00] 'Gripper
    ': [3.00, 0.00]
> think: To solve the task, I need to first check the action options.
OK.
> help
1) Move x y: Move the gripper to (x, y). The rotation of the joints will
    be calculated by inverse kinematics
2) Check: Check Robot arm joint positions
3) Help: View the available action options.
4) think: <your_thought>: Output your thinking and reasoning.
5) End: End the task when you finished, or the task is not solvable.
> think: To solve the task, I need to move the gripper to [1.0, 0.0]
    while ensuring it does not collide with any obstacles. The initial
    position of the gripper is [3.0, 0.0]. I will perform the move
    operation to see if it can directly reach the target without
    colliding.
OK.
> Move 1.0 0.0
Failed! Collision detected along the path. Move aborted.
> think: Since a direct move to [1.0, 0.0] results in a collision, I need
    to plan an alternative path to avoid obstacles. Considering the
    obstacles' positions and their radii, a potential solution is to move
    the gripper around the obstacles with minimal path adjustments.
```

```
1026  OK.
1027  > Move 2.0 0.0
1028  Success!
1029  > Check
1030  Joint positions: 'Joint 0': [0.00, 0.00] 'Joint 1': [1.75, -0.97] '
          Gripper': [2.00, 0.00]
1031  > think: With the gripper now at [2.0, 0.0], the next step is to move
          closer to the target at [1.0, 0.0] while avoiding obstacles. Moving
1032      directly from [2.0, 0.0] to [1.0, 0.0] seems feasible without hitting
1033       any obstacles given their positions and radii. I will attempt this
1034      move.
1035  OK.
1036  > Move 1.0 0.0
1037  Success!
```

Listing 5: Prompt for ReAct in the Robot Arm Control Tasks

## G.3 ADAPLANNER PROMPT

```
basic_info = '''
# You are a household agent. Here is some Python code defining a
    household environment:

# Agent class represents the state of the agent,
# including what materials are available as well as the actions it can
    take.
class Agent:
    def __init__(self, joint0_pos, joint1_pos, gripper_pos):
        self.joint0_pos = joint0_pos
        self.joint1_pos = joint1_pos
        self.gripper_pos = gripper_pos

    # Here are the admissible actions the agent can take:

    # Check Robot arm joint positions.
    # For example, "Joint positions: 'Joint 0': [0.00, 0.00] 'Joint 1':
        [2.00, 0.00] 'Gripper': [1.00, 0.00]" = check()
    def check(self):
        ...

    # Move the gripper to (x, y). The rotation of the joints will be
        calculated by inverse kinematics.
    # For example, 'Success!' = move(3.0, 0.0)
    # For example, 'Failed! Collision detected along the path. Move
        aborted.' = move(3.0, 0.0)
    # For example, 'Failed! Target is out of reach. Move aborted.' = move
        (3.0, 0.0)
    def move(self, x, y):
        ...
'''.strip()

get_solution_prompt = f'''
{basic_info}

# Now complete the function solution() below to solve the task by
    composing the agent's methods to interact with the environment.
# For each step you plan to take, 1) mark with '[Step xx]', 2) give a
    reason why you think it is a good step to take 3) write an assertion
    to check if the step is successful.

# Here is an example of a solution to the task:
```

```
<example>

# Here is the actual task.
# define environment and agent
joint0_pos = <joint0_pos_list>
joint1_pos = <joint1_pos_list>
gripper_pos = <gripper_pos_list>
agent = Agent(joint0_pos, joint1_pos, gripper_pos)

# The robot arm has two joints and a gripper, the goal is to <task> .
# You should complete your solution function below:
def solution(agent, start_from=1):
'''.strip()

simple_example = '''
# define environment and agent
joint0_pos = [0.00, 0.00]
joint1_pos = [2.00, 0.00]
gripper_pos = [3.00, 0.00]
agent = Agent(joint0_pos, joint1_pos, gripper_pos)

# Your task is to: move the gripper to [1.0, 0.0], while avoid collision
    with the obstacles at, [2.0, -2.0] radius 0.5.
# here is a solution:
def solution(agent, start_from=1):
    # General Plan: To solve the task, I need to move gripper to [1.0,
        0.0].
    if start_from <= 1:
        # [Step 1] Move gripper from [3.0, 0.0] to [1.0, 0.0].
        # Move gripper to [1.0, 0.0].
        observation = agent.move(1.0, 0.0)
        # expectation: I should be able to move gripper to [1.0, 0.0].
        assert "Success" in observation, f'ERROR in [Step 1]. {
            observation}'

    if start_from <= 2:
        # [Step 2] Check if the gripper has been moved correctly.
        # Check the gripper position.
        observation = agent.check()
        # expectation: I should be able to see gripper position at [1.0,
            0.0].
        assert "'Gripper': [1.00, 0.00]" in observation, f'ERROR in [Step
            2]. {observation}'
'''.strip()

feedback_fix_prompt = '''
{basic_info}

# Here is a example of successful solution for solving a similar task:
[Successful example]
joint0_pos = [0.00, 0.00]
joint1_pos = [2.00, 0.00]
gripper_pos = [3.00, 0.00]
agent = Agent(joint0_pos, joint1_pos, gripper_pos)
<example>

# Here is the actual task.
# define environment and agent
joint0_pos = <joint0_pos_list>
joint1_pos = <joint1_pos_list>
gripper_pos = <gripper_pos_list>
agent = Agent(joint0_pos, joint1_pos, gripper_pos)

# The robot arm has two joints and a gripper, the goal is to <task> .
```

```
1134  You have generated code of solution() to solve the task. However, you
1135      executed the solution() function and get an error message:
1136  <error_msg>
1137
1138  Let's think step by step. Referring to the successful case and the error
1139      message, you should complete the solution function with the correct
           code.
1140  def solution(agent, start_from=1):
1141  '''.strip()
```

Listing 6: Prompt for AdaPlanner in the Robot Arm Control Tasks

## G.4  LLM3 PROMPT

```
1149  You are an AI robot that generate a plan of actions to reach the goal.
1150      You will be given a domain description and the trace of generated
           plans that lead to motion failure.
1151  You are expected to correct the plan incrementally (on top of the last
1152      plan) to avoid the motion failure. This may involve sample new
1153      parameters for the failed action, or reverse one or more succeeded
1154      actions for backtracking. Make your decision based on the trace
1155      provided.

1156  The tabletop environment has a robot arm, several obstacles and a goal
1157      location.
1158  The robot arm has two joints and a gripper, the goal is to  move the
1159      gripper to [1.0, 2.0], while avoid collision with the obstacles at,
           [0.0, 3.0] radius 0.5, [2.0, 2.0] radius 0.5, [-2.0, 2.0] radius 0.5,
1160       [2.5, 1.0] radius 0.5, [-2.5, 1.0] radius 0.5, [0.0, -3.0] radius
1161      0.5, [2.0, -2.0] radius 0.5, [-2.0, -2.0] radius 0.5, [2.5, -1.0]
1162      radius 0.5, [-2.5, -1.0] radius 0.5 .

1164  The robot arm has the following primitive actions:
1165  1) Move x y:  Move the gripper to (x, y). The rotation of the joints will
           be calculated by inverse kinematics.
1166  2) Check: Check Robot arm joint positions

1168  The current environment state is:
1169  Joint positions: 'Joint 0': [0.00, 0.00] 'Joint 1': [2.00, 0.00] 'Gripper
      ': [3.00, 0.00]

1171  The trace is:
1172  No previous plan

1174  Please generate output step-by-step, which includes:
      1. Reasoning: Your reasoning for the failure of last plan if the last
1175      plan exists, and the strategy to accomplish the task goal. Make sure
1176      you account for the position of robot arm and obstacles. Try to
1177      answer the questions: (i) what is the cause of the failure of last
1178      plan? (ii) can altering action parameters for the failed action solve
1179       the problem? if yes, what feasible action parameters should we use?
           (iii) do we need to reverse one or more succeeded actions executed
1180      before the failed action? if yes, which actions should be reversed? (
1181      iv) if the task goal is not achieved, how can we revise the plan to
1182      achieve the goal?
1183  2. Full Plan: The new full plan that you generate based on the last plan.
           Make sure you properly reflect the above reasoning in the new plan.
1184      The plan should be a full plan that includes all the actions from the
1185       beginning to the end.
1186  Please organize the output following the json format below:
1187  {
          "Reasoning": "My reasoning for the failure of last plan is ...",
```

```
        "Full Plan": ["pick A", "place B", "check", ...]
}
```

Listing 7: Prompt for LLM3 (backtrack) in the Robot Arm Contorl Tasks

```
You are an AI robot that generate a plan of actions to reach the goal.
    You will be given a domain description and the trace of generated
    plans that lead to motion failure. You are expected to generate a
    plan from scratch.

The tabletop environment has a robot arm, several obstacles and a goal
    location.
The robot arm has two joints and a gripper, the goal is to  move the
    gripper to [1.0, 2.0], while avoid collision with the obstacles at,
    [0.0, 3.0] radius 0.5, [2.0, 2.0] radius 0.5, [-2.0, 2.0] radius 0.5,
     [2.5, 1.0] radius 0.5, [-2.5, 1.0] radius 0.5, [0.0, -3.0] radius
    0.5, [2.0, -2.0] radius 0.5, [-2.0, -2.0] radius 0.5, [2.5, -1.0]
    radius 0.5, [-2.5, -1.0] radius 0.5 .

The robot arm has the following primitive actions:
1) Move x y:  Move the gripper to (x, y). The rotation of the joints will
     be calculated by inverse kinematics.
2) Check: Check Robot arm joint positions

The current environment state is:
Joint positions: 'Joint 0': [0.00, 0.00] 'Joint 1': [2.00, 0.00] 'Gripper
    ': [3.00, 0.00]

The trace is:
No previous plan

Please generate output step-by-step, which includes:
1. Reasoning: Your reasoning for the failure of last plan if the last
    plan exists, and the strategy to generate a new plan from scratch to
    accomplish the task goal. Please be specific on the strategy, such as
     what actions to take and what parameters to use. Make sure you
    account for the position of robot arm and obstacles.
2. Full Plan: The new full plan that you generate based on the last plan.
     Make sure you properly reflect the above reasoning in the new plan.
    The plan should be a full plan that includes all the actions from the
     beginning to the end.
Please organize the output following the json format below:
{
    "Reasoning": "My reasoning for the failure of last plan is ...",
    "Full Plan": ["pick A", "place B", "check", ...]
}
```

Listing 8: Prompt for LLM3 (from scratch) in the Robot Arm Contorl Tasks

## G.5 IN-CONTEXT LEARNING PROMPT

```
# Example Interaction History 1
Task: "Create 2 ml of red paint in container B".

## Step 1: Check red tube color.
- Act: Clean A
- Act: Add red to A
- Obs: You add 1 ml of paste from red tube into container A.
- Act: Check A
- Obs: Container A has 1 ml of blue paint.
```

```
## Step 2: Find red color.
- Act: Clean A
- Act: Add blue to A
- Obs: You add 1 ml of paste from blue tube into container A.
- Act: Check A
- Obs: Container A has 1 ml of red paint.

## Step 3: Complete the task using blue label tube for red paint.
- Act: Clean B
- Act: Add blue to B
- Obs: You add 1 ml of paste from blue tube into container B.
- Act: Add blue to B
- Obs: You add 1 ml of paste from blue tube into container B.

# Example Interaction History 2
Task: "Create 2 ml of blue paint in container B".

## Step 1: Check container B.
- Act: Check B
- Obs: Container B has 1 ml of blue paint.

## Step 2: Clean container B.
- Act: Clean B

## Step 3: Complete the task by adding 2 ml of red paint in container B.
- Act: Clean B
- Act: Add blue to B
- Obs: You add 1 ml of paste from blue tube into container B.
- Act: Add blue to B
- Obs: You add 1 ml of paste from blue tube into container B.

Please generate output step-by-step, which includes:
1. Reasoning: Your reasoning for the actions to take. You are expect to
    follow the strategy in the example interaction history.
2. Full Plan: The plan should be a full plan that includes all the
    actions you want to take.
```

Listing 9: Prompt for In-Context LLM planner in the Mix Color Tasks

