# OpenReview forum: "Information Seeking for Robust Decision Making under Partial Observability"
_ICLR.cc/2026/Conference — Submitted to ICLR 2026_

### Official Review · Reviewer_TXuA · 2025-10-28

**Soundness:** 2
**Presentation:** 3
**Contribution:** 2
**Rating:** 4
**Confidence:** 4

**Summary:**

Existing large language model (LLM)-based planners for agents in partially observable environments have limitations. These agents often ignore the mismatch between their internal dynamics and the actual environment, leading to decision biases. To address this issue, the paper proposes an Information Seeking Decision Planner (InfoSeeker), an LLM-based framework that combines task-oriented planning with information seeking. InfoSeeker prompts the LLM to actively gather information through planned actions before generating or modifying task plans, ensuring alignment between internal dynamics and the real-world situation. The authors also introduce a new benchmark dataset to test agent performance under conditions of observation uncertainty and environmental noise. Experimental results show that InfoSeeker outperforms baseline methods on this new benchmark, validating the effectiveness of the proposed approach.

**Strengths:**

1、Combination of Task Planning and Information Seeking:
Although the innovation is not strong, the framework presents the idea of incorporating information seeking into task planning. Theoretically, it offers an approach for solving some partially observable problems.

2、Cross-Model Applicability:
InfoSeeker can be tested across different LLM models, showing a certain level of generality.

**Weaknesses:**

1、Simplicity of the Framework:
The framework is quite simple both theoretically and in implementation. It lacks deeper innovation or the ability to solve more complex problems.

2、Lack of Depth in Experimental Design:
The experiments are limited to simple benchmark tasks, and all tasks and datasets were designed by the authors themselves, which makes the validation of its general applicability insufficient.

3、Lack of Real-World Validation:
The experimental results are primarily based on simulated environments (and in pure text), which cannot verify the actual effectiveness of the framework in real-world, complex environments.

4、Issues with the Information Extraction Module:
The deficiencies in the information extraction module led to some failures in the experiments, and the paper does not propose effective solutions to address this issue.

5、Low Final Pass Rate on the TravelPlanner Benchmark:
On the TravelPlanner benchmark, all methods show very low final pass rates. Although InfoSeeker performs the best, the absolute improvement is too small to convincingly demonstrate its significant advantage in real, complex tasks.

**Questions:**

1、How can the framework in the paper be applied in more complex and dynamic real-world environments? Will it perform well in highly practical tasks?

2、Does the design of the custom dataset affect the generalizability of the results? Can this method demonstrate its advantages on other public datasets? Please add further experiments.

3、Regarding the failures in the information extraction module, are there clear optimization solutions, or have other technological approaches been considered to improve its performance?

---

> ### Author Response · Authors · 2025-11-18
> **Rebuttal by Authors**
>
> We thank the reviewer for their suggestions and address their concerns below.
> ## Weakness 1
> > Simplicity of the Framework: The framework is quite simple both theoretically and in implementation. It lacks deeper innovation or the ability to solve more complex problems.
>
> Existing LLM agents revise their plans based solely on past execution traces. In contrast, InfoSeeker actively performs information-seeking actions to gather observations to align its internal dynamics with the environment before planning (see Section 1 and Figure 1). As shown in Section 5, InfoSeeker achieves state-of-the-art performance on our benchmark as well as on two additional, well-established public benchmarks that involve practical and complex problem settings.
>
> ## Weakness 2 & 3 and Questions 1 & 2
> >The experiments are limited to simple benchmark tasks, and all tasks and datasets were designed by the authors themselves ... .
> >Lack of Real-World Validation ... which cannot verify the actual effectiveness of the framework in real-world, complex environments.
>
> >How can the framework in the paper be applied in more complex and dynamic real-world environments? Will it perform well in highly practical tasks?
> >Does the design of the custom dataset affect the generalizability of the results? Can this method demonstrate its advantages on other public datasets? Please add further experiments.
>
> Our paper evaluated InfoSeeker on three benchmarks: one designed by us and two well-established public benchmarks, LLM3 and TravelPlanner. LLM3 tests robot arm manipulation in a physical simulator, while TravelPlanner assesses the use of internet tools for practical travel planning tasks. As shown in Section 5.2, InfoSeeker achieved state-of-the-art performance across these diverse settings, demonstrating its effectiveness and adaptability for complex, real-world tasks.
>
> ## Weakness 4
> > Issues with the Information Extraction Module: The deficiencies in the information extraction module led to some failures in the experiments, and the paper does not propose effective solutions to address this issue.
>
> Enhancing LLMs’ summarization capabilities is beyond the scope of this work on LLM agents. We had discussed this limitation in Section 7 and noted it as an important avenue for future research.
>
> ## Weakness 5
> > Low Final Pass Rate on the TravelPlanner Benchmark: On the TravelPlanner benchmark, all methods show very low final pass rates. Although InfoSeeker performs the best, the absolute improvement is too small to convincingly demonstrate its significant advantage in real, complex tasks.
>
> TravelPlanner is a challenging benchmark, with previous state-of-the-art results showing very low final pass rates (only 4.44). InfoSeeker achieves state-of-the-art performance on TravelPlanner and also demonstrates strong performance gain on both LLM3 and our own proposed benchmark. These results highlight InfoSeeker’s ability to generalize effectively across a diverse set of real-world tasks.
>
> ## Question 3
> >Regarding the failures in the information extraction module, are there clear optimization solutions, or have other technological approaches been considered to improve its performance?
>
> Enhancing LLMs’ summarization capabilities is beyond the scope of this work on LLM agents. One possible improvement is to use LLMs with advanced reasoning capabilities to improve the information extraction module. How to improve LLMs’ summarization capabilities remains an interesting direction for future research.

---

> > ### Author Response · Authors · 2025-11-24
> > **Happy to have further discussion**
> >
> > Dear Reviewer TXuA,
> >
> > Thank you for reviewing our paper. We appreciate your valuable feedback and have carefully posted our responses. We understand the reviewers may have limited time, but we are happy to provide further clarification or engage in discussion during the Author–Reviewer Discussion phase. Looking forward to your feedback!
> >
> > Best,
> >
> > Authors of submission 14434

---

### Official Review · Reviewer_WjYP · 2025-10-29

**Soundness:** 3
**Presentation:** 3
**Contribution:** 2
**Rating:** 4
**Confidence:** 4

**Summary:**

To address the problem of LLM-based planners in interactive tasks with uncertain outcomes, the authors propose the Information Seeking Decision Planner (InfoSeeker), an LLM decision-making framework that integrates task-oriented planning with information seeking. It proactively obtains environmental information alongside with execution history, and then revises its execution plan based on updated beliefs (world knowledge). Through experiments on multiple self-made benchmarks and two recent public benchmarks, the authors demonstrate that the proposed method is promising.

**Strengths:**

1. Dealing with uncertainty in LLM-based planning is an important and less-explored topic.
2. The proposed method is effective.
3. The paper is well-written.

**Weaknesses:**

1. Limited novelty, it seems that such a method is a combination of ReAct + Reflexion.
2. The baselines are incomplete. For example, ReAct+Reflexion, Tree of Thought, Self-consistency, or MCTS-based planning method can be used as the baselines.

**Questions:**

1. What are the token consumptions?
2. What are the differences between InfoSeeker and LLM3?
3. Some related works are missing. Please perform a more comprehensive survey regarding LLM and robust/uncertainty decision making/reasoning.

---

> ### Author Response · Authors · 2025-11-18
> **Rebuttal by Authors**
>
> We thank the reviewer for their constructive and thoughtful suggestions and address their concerns below.
> ## Weakness 1
> >Limited novelty, it seems that such a method is a combination of ReAct + Reflexion.
>
> ReAct + Reflexion revises its plans based solely on summaries of past execution traces. In contrast, InfoSeeker actively performs information-seeking actions to gather informative observations that are critical for aligning its internal dynamics with the environment before generating each new plan.  As shown in the table below, our approach doubles the success rates while consuming less than one-third of the tokens required by ReAct + Reflexion on the robot arm control and stack single block tasks.
>
> | Success Rate (Token Consumption) | Robot Arm-Basic | Robot Arm-Perturbed | Stack Single Block-Basic | Stack Single Block-Perturbed |
> | :------------------------- | :-------------- | :------------------ | :----------------------- | :--------------------------- |
> | ReAct + Reflexion          | 24% (73k)       | 16% (71k)           | 70% (81k)                | 34% (103k)                   |
> | COT-Self Consistency       | 10% (93k)       | 14% (94k)           | **86%** (118k)           | 4% (48k)                     |
> | Tree-Of-Thoughts           | 30% (92k)       | 34% (91k)           | **86%** (12k)            | 2% (46k)                     |
> | InfoSeeker (ours)          | **100%** (20k)  | **80%** (22k)       | 82% (9k)                 | **62%** (13k)                |
>
> ## Weakness 2
> >The baselines are incomplete. For example, ReAct+Reflexion, Tree of Thought, Self-consistency, or MCTS-based planning method can be used as the baselines.
>
> Compared with the baselines suggested by reviewer, InfoSeeker not only achieves the highest success rates but also consumes far fewer tokens (shown in the table above), demonstrating both superior performance and efficiency.
>
> ## Question 1
> >What are the token consumptions?
>
> As shown in the table below, InfoSeeker improves success rates substantially while maintaining token consumption comparable to the LLM3 baselines.
>
> | Success Rate (Token Consumption) | Robot Arm-Basic | Robot Arm-Perturbed | Stack Single Block-Basic | Stack Single Block-Perturbed |
> | :------------------------- | :-------------- | :------------------ | :----------------------- | :--------------------------- |
> | ReAct                      | 48% (18k)       | 2% (19k)            | 68% (139k)               | 22% (100k)                   |
> | AdaPlanner                 | 12% (39k)       | 0% (39k)            | 14% (323k)               | 0% (349k)                    |
> | LLM3 (backtrack)           | **100%** (87k)  | 6% (26k)            | 76% (3k)                 | 6% (13k)                     |
> | LLM3 (from scratch)        | 98% (9k)        | 2% (28k)            | **86%** (2k)             | 0% (12k)                     |
> | InfoSeeker (ours)          | **100%** (20k)  | **80%** (22k)       | 82% (8k)                 | **62%** (13k)                |
>
> ## Question 2
> >What are the differences between InfoSeeker and LLM3?
>
> Similar to ReAct + Reflexion, LLM3 revises its plans based solely on past execution traces. In contrast, InfoSeeker actively performs information-seeking actions to gather observations and aligns its internal dynamics with the environment before planning. As shown in the table above, InfoSeeker achieves substantially higher success rates with minimal additional token usage relative to the LLM3 baselines.
>
> ## Question 3
> >Some related works are missing. Please perform a more comprehensive survey regarding LLM and robust/uncertainty decision making/reasoning.
>
> We will update the paper to include new results for the suggested baselines, as well as a more extensive survey of LLM reasoning methods.

---

> > ### Author Response · Authors · 2025-11-24
> > **Happy to have further discussion**
> >
> > Dear Reviewer WjYP,
> >
> > Thank you for reviewing our paper. We appreciate your valuable feedback and have carefully posted our responses. We understand the reviewers may have limited time, but we are happy to provide further clarification or engage in discussion during the Author–Reviewer Discussion phase. Looking forward to your feedback!
> >
> > Best,
> >
> > Authors of submission 14434

---

### Official Review · Reviewer_t7hd · 2025-10-30

**Soundness:** 3
**Presentation:** 3
**Contribution:** 2
**Rating:** 6
**Confidence:** 4

**Summary:**

The paper proposes InfoSeeker, an LLM-based decision-making approach in partially observable settings that integrates information seeking actions to align the LLM-agent’s belief of the dynamics with that of the environment for effective task-oriented planning. Information seeking is done by running diagnostic trials and observing the outcomes to detect shifts in environment dynamics. To evaluate their approach, the paper also presents text-based benchmarks that include uncertain dynamics in tasks such as robot arm control and navigation, color mixing, and block stacking based on existing benchmarks.

**Strengths:**

- The paper is generally well-written and clear.
- The contributions of the paper in the context of existing works are communicated clearly. Specifically, the paper adds information seeking as an explicit step before generating/refining the plans as opposed to prior works that rely solely on reactive feedback.
- The paper additionally focuses on dealing with uncertainty in environment dynamics, in addition to observation uncertainty considered in prior works.
- The formalization of the problem in terms of the POMDP framework is meaningful.
- Experiments and related ablation studies are promising. Failure analysis is helpful to readers.

**Weaknesses:**

- The paper tackles only a very specific kind of uncertainty in dynamics: those that are deterministic but unknown/unmodeled beforehand (e.g. deterministic shift in gripper pose due to misaligned robot arm). Many problems of interest in the broader planning under uncertainty domain are more focused on uncertainty due to stochasticity (e.g. stochastic action outcomes due to imperfect/noisy actuators).
- There are few places in the paper where the authors claim their approach to make “optimal” decisions under uncertainty. Perhaps, “effective” might be a better word for it, since “optimal” is a little strong given that in some experiments, the paper’s approach underperforms existing baselines. Claiming optimality might also call for establishing formal guarantees that no other approach outperforms the paper’s approach. This might come off as misleading especially since the presented approach is theoretically formalized in the context of the standard POMDP framework, which often has clear definitions of optimality.
- The proposed benchmark is limited and feels hand-crafted as acknowledged by the authors.

**Questions:**

- In line with the first weakness mentioned above, what are the limitations and potentials of the proposed approach for dealing with the uncertainty due to stochasticity in addition to those due to unknown/unmodeled dynamics?
- Given that the proposed benchmarks are limited in scope (acknowledged in the limitations), what are the challenges and limitations of the proposed approach for applications in more realistic planning scenarios in the real world (e.g. with an embodied AI agent)?

---

> ### Author Response · Authors · 2025-11-18
> **Rebuttal by Authors**
>
> We thank the reviewer for their constructive and thoughtful suggestions and address their concerns below.
> ## Weakness 1 and Question 1
> > The paper tackles only a very specific kind of uncertainty in dynamics: those that are deterministic but unknown/unmodeled beforehand. Many problems of interest in the broader planning under uncertainty domain are more focused on uncertainty due to stochasticity. What are the limitations and potentials of the proposed approach for dealing with the uncertainty due to stochasticity in addition to those due to unknown/unmodeled dynamics?
>
> Yes, you’re right, our proposed method does not generalize to stochastic dynamics, as existing LLMs are not trained to model such environments (Qiu et al., 2025). Developing methods to enable LLMs to model stochastic dynamics by leveraging the Bayesian framework represents an interesting direction for future work.
>
> **Reference:**
> Qiu, et al. (2025). *Bayesian teaching enables probabilistic reasoning in large language models.* arXiv preprint arXiv:2503.17523.
> ## Weakness 2
> >There are few places in the paper where the authors claim their approach to make “optimal” decisions under uncertainty. ... This might come off as misleading especially since the presented approach is theoretically formalized in the context of the standard POMDP framework, which often has clear definitions of optimality.
>
> Thanks for the advice! We will revise the wording by replacing “optimal” with “effective” in the camera-ready version to avoid confusion.
>
> ## Weakness 3
> > The proposed benchmark is limited and feels hand-crafted as acknowledged by the authors.
>
> Through this paper, we aim to present the challenge of learning-free adaptation in partially observable environments with uncertain dynamics. We hope our paper can encourage the development of more extensive and rigorous benchmarks to advance research in this underexplored area of LLM planning.
>
> ## Question 2
> > Given that the proposed benchmarks are limited in scope (acknowledged in the limitations), what are the challenges and limitations of the proposed approach for applications in more realistic planning scenarios in the real world (e.g. with an embodied AI agent)?
>
> Current LLMs cannot handle continuous dynamics in real-world planning scenarios. Exploring ways to integrate them with low-level continuous control is an important direction for future research in planning for embodied agents. We will add this limitation to our paper.

---

> > ### Author Response · Authors · 2025-11-24
> > **Happy to have further discussion**
> >
> > Dear Reviewer t7hd,
> >
> > Thank you for reviewing our paper. We appreciate your valuable feedback and have carefully posted our responses. We understand the reviewers may have limited time, but we are happy to provide further clarification or engage in discussion during the Author–Reviewer Discussion phase. Looking forward to your feedback!
> >
> > Best,
> >
> > Authors of submission 14434

---

### Official Review · Reviewer_gTmW · 2025-11-01

**Soundness:** 2
**Presentation:** 2
**Contribution:** 2
**Rating:** 4
**Confidence:** 3

**Summary:**

The paper introduces an LLM-based decision-making framework that interleaves task‑oriented planning with explicit information‑seeking actions. The goal is to align the agent's internal dynamics with the actual environment, particularly under partial observability and noisy or shifted dynamics. The method prompts the LLM to diagnose inconsistencies, design small diagnostic trials, extract insights from those trials, and then revise task plans. The authors also propose a text-based benchmark that injects uncertainty not only in observations but also in environmental dynamics. Empirically, it reportedly achieves large performance gains on the new benchmark and shows improvements on LLM3-style robotic manipulation and TravelPlanner web navigation.

**Strengths:**

1. The work moves beyond observation uncertainty to uncertain/shifted dynamics, a critical but under‑evaluated challenge for LLM agents.

2. The work shows a practical, training‑free loop and the cycle is simple to adopt and demonstrably helpful across tasks.

**Weaknesses:**

1. The claimed formal connection to POMDPs remains interpretive. There is no explicit specification of the belief representation, the backup operator the prompts approximate, or a principled value‑of‑information objective that the information‑seeking step maximizes.

2. The writing is not very clear. For example, the text refers to internal dynamics, it is unclear what gets updated (offsets? action‑effect map? parameters), where it is stored (text memory? structured map?), and how it is consumed during planning.

3. There is no clear trigger for when to seek information, no seek budget, and no rule for choosing diagnostic actions. It is very underspecified.

4. 4.1 Talks about 5 tasks, and yet only 4 are explained (I suppose the last one is split into 2, but it does not make it a new task). And in 5.1, what are 11 tasks from the benchmark? The writing is very hard to follow.

5. There is an inconsistent mixing of steps, attempts, and wall‑clock time across sections without normalization, weakening sample‑efficiency claims.

6. Adding to (5): Efficiency is multi-dimensional (env steps, LLM calls, tokens, time). Claims are based partly on a wall‑clock comparison. But, token/LLM‑call budgets and env‑step budgets are not jointly reported.

7. Similarly, isn't a "one‑minute" cap API‑latency‑ and hardware‑dependent? It is not a stable cross‑paper comparator.

8. It is unclear whether baselines had identical total interaction budgets (seek+plan vs plan‑only), identical temperature/top‑p, timeouts, and truncation policies.

9. Only two LLMs are tested and per‑model variability and versioning are not reported. I would strongly suggest the authors to ablate a few more LLMs.

**Questions:**

Please see the weaknesses section.

---

> ### Author Response · Authors · 2025-11-18
> **Rebuttal by Authors**
>
> We thank the reviewer for their constructive and thoughtful suggestions and address their concerns below.
>
> ## Weakness 1
> > The claimed formal connection to POMDPs remains interpretive. There is no explicit specification of the belief representation, the backup operator the prompts approximate, or a principled value‑of‑information objective that the information‑seeking step maximizes.
>
> We like to clarify that our work is not a theoretical paper. Our focus is on proposing an LLM agent that can achieve learning-free, zero-shot generalization to unseen tasks. LLM agents do not require an explicit belief representation or the learning of a value function to update their internal dynamics. Integrating LLM agents with reinforcement learning–based approaches is an interesting direction for future research.
>
> ## Weakness 2
> > The writing is not very clear. For example, the text refers to internal dynamics, it is unclear what gets updated (offsets? action‑effect map? parameters), where it is stored (text memory? structured map?), and how it is consumed during planning.
>
> Following prior works (Sawyer et al., 2024), InfoSeeker applies the LLM’s internal reasoning to update its internal dynamics, and generate plans based on the updated dynamics. These dynamics are not retained after execution.
>
> **Reference:**
> Sawyer, et al. (2024). *Can foundation models actively gather information in interactive environments to test hypotheses?* arXiv preprint arXiv:2412.06438.
>
> ## Weakness 3
> > There is no clear trigger for when to seek information, no seek budget, and no rule for choosing diagnostic actions. It is very underspecified.
>
> Each task-oriented planning phase is preceded by an information-seeking phase. During this phase, it decides which actions to gather information for, and how many, based on its internal reasoning.
>
> ## Weakness 4
> > 4.1 Talks about 5 tasks, and yet only 4 are explained. And in 5.1, what are 11 tasks from the benchmark?
>
> We thank the reviewer for pointing out the ambiguity. The benchmark contains five core tasks:
> 1. Robot arm control
> 2. Robot navigation
> 3. Mix colors
> 4. Stack single block
> 5. Stack multiple blocks
>
> Each core task has a basic and a perturbed variants, except for color mixing, which has one basic and two perturbed variants. This results in a total of 11 task variants. We will update Sections 4.1 and 5.1 to clarify this structure and prevent any misunderstanding.
>
> ## Weakness 5
> > There is an inconsistent mixing of steps, attempts, and wall‑clock time across sections without normalization, weakening sample‑efficiency claims.
>
> We follow standard RL practice by reporting success rate under a fixed environment-step budget. In contrast to typical RL policies, LLMs incur additional inference time and computational cost for reasoning and plan generation. To facilitate practical use, we also account for these overheads by reporting success rate with respect to wall-clock time and to different numbers of planning attempts.
>
> ## Weakness 6
> > Efficiency is multi-dimensional (env steps, LLM calls, tokens, time). Claims are based partly on a wall‑clock comparison. But, token/LLM‑call budgets and env‑step budgets are not jointly reported.
>
> Our main experimental results, reported in Table 1 and discussed in Section 5.1, evaluate task success rates under a budget of 100 environment steps. The tables below show the average number of tokens and LLM calls per task for the same set of experiments. InfoSeeker outperforms all baselines by a substantial margin while using a similar number of tokens as the two LLM3 variants, demonstrating both superior effectiveness and efficiency.
>
> | Robot Arm Control – Basic | Success Rate | LLM Calls | Token Usage |
> | :--- | :---: | ---: | ---: |
> | ReAct | 48 | 16.94 | 18400.56 |
> | AdaPlanner | 12 | 14.00 | 39316.90 |
> | LLM3 (backtrack) | **100** | 5.98 | 8665.62 |
> | LLM3 (from scratch) | 98 | 5.86 | 9072.48 |
> | InfoSeeker (ours) | **100** | 16.98 | 20292.36 |
>
> | Robot Arm Control – Perturbed | Success Rate | LLM Calls | Token Usage |
> | :--- | :---: | ---: | ---: |
> | ReAct | 2 | 17.50 | 19041.56 |
> | AdaPlanner | 0 | 14.00 | 38752.82 |
> | LLM3 (backtrack) | 6 | 15.86 | 25933.14 |
> | LLM3 (from scratch) | 2 | 16.50 | 28271.46 |
> | InfoSeeker (ours) | **80** | 18.86 | 22099.24 |
>
> | Stack Single Block – Basic | Success Rate | LLM Calls | Token Usage |
> | :--- | :---: | ---: | ---: |
> | ReAct | 68 | 62.32 | 139445.14 |
> | AdaPlanner | 14 | 89.68 | 323878.70 |
> | LLM3 (backtrack) | 76 | 2.16 | 3480.86 |
> | LLM3 (from scratch) | **86** | 2.18 | 2209.92 |
> | InfoSeeker (ours) | 82 | 7.32 | 8879.98 |
>
> | Stack Single Block – Perturbed | Success Rate | LLM Calls | Token Usage |
> | :--- | :---: | ---: | ---: |
> | ReAct | 22 | 47.08 | 100219.30 |
> | AdaPlanner | 0 | 102.00 | 349688.62 |
> | LLM3 (backtrack) | 6 | 6.50 | 12596.74 |
> | LLM3 (from scratch) | 0 | 6.40 | 11928.06 |
> | InfoSeeker (ours) | **62** | 10.46 | 13159.52 |

---

> ### Author Response · Authors · 2025-11-18
> **Rebuttal by Authors - Continued**
>
> ## Weakness 7
> >Similarly, isn't a "one‑minute" cap API‑latency‑ and hardware‑dependent? It is not a stable cross‑paper comparator.
>
> All experiments in our paper are conducted on the same machine and use the same official APIs from OpenAI and Google.
>
> ## Weakness 8
> >It is unclear whether baselines had identical total interaction budgets (seek+plan vs plan‑only), identical temperature/top‑p, timeouts, and truncation policies.
>
> In Table 1 all baseline methods and our proposed InfoSeeker are evaluated under a budget of 100 environment steps per task. Additionally, the left panel of Figure 3 reports the success rates of seek-and-plan and plan-only methods across different environment step budgets. All experiments in our paper use identical temperature, top-p, timeout, and truncation policies, following the default configurations provided by the official OpenAI and Google APIs. We will update our paper to clarify the experimental setup.
>
> ## Weakness 9
> > Only two LLMs are tested and per‑model variability and versioning are not reported. I would strongly suggest the authors to ablate a few more LLMs.
>
> Thanks for the suggestion! We additionally conducted experiments on robot arm control using a more recent Language-Reasoning Model–OpenAI o4-mini (released on 2025/04/16).  With this up-to-date model, InfoSeeker still consistently outperformed all other baselines, demonstrating InforSeeker’s robustness to different LLMs. We will update the paper with our new results on up-to-date LLMs, and list the specific versions of each LLMs.
>
> | o4 mini | Robot Arm-Basic | Robot Arm-Perturbed |
> | :---- | :---- | :---- |
> | AdaPlanner | 0 | 0 |
> | LLM3 BT | 64 | 38 |
> | LLM3 FS | 20 | 34 |
> | InfoSeeker (ours) | **70** | **64** |

---

> > ### Author Response · Authors · 2025-11-24
> > **Happy to have further discussion**
> >
> > Dear Reviewer gTmW,
> >
> > Thank you for reviewing our paper. We appreciate your valuable feedback and have carefully posted our responses. We understand the reviewers may have limited time, but we are happy to provide further clarification or engage in discussion during the Author–Reviewer Discussion phase. Looking forward to your feedback!
> >
> > Best,
> >
> > Authors of submission 14434

---

> > ### Comment · Reviewer_gTmW · 2025-11-24
> >
> > Dear authors, thank you for your detailed rebuttal (and all the work), I would like to make a few quick points: w.r.t. W1, I would like to point out that, one of the key contributions (as mentioned in the paper) is (contribution 3): A formal connection between LLM-based planning and Partially Observable Markov Decision Processes (POMDPs). I get that this is not a theoretical paper, so I would suggest rephrasing the contribution. I would also suggest that the authors update the writing in the paper to fix W2.
> >
> > That being said, I am happy with the response and it answers all the questions I had. I would like to move my rating up (from 4 to 6).
> >
> > - Reviewer gTmW

---

> > > ### Author Response · Authors · 2025-11-24
> > >
> > > Thank you for the feedback! We will revise Contribution 3 and update the writing to address W2 in the camera-ready version.

---

> > > > ### Author Response · Authors · 2025-12-03
> > > > **Additional Experiments Addressing Weakness 9**
> > > >
> > > > We repeated the Table 1 experiments using the latest Gemini Flash model, gemini-2.5-flash (updated June 2025). With this newer model, InfoSeeker achieved **over 60% absolute performance gains** on the perturbed variants of Robot Arm Control and Robot Navigation, further demonstrating its robustness across different LLMs. We will update the paper to include these new results obtained with o4-mini and gemini-2.5-flash.
> > > >
> > > > | Gemini 2.5 Flash       | Robot Arm Control-Basic | Robot Arm Control-Perturbed | Mix Color-Basic | Mix Color-Contaminated | Mix Color-Wronglabel |
> > > > | :---------------------- | :---------------------- | :-------------------------- | :-------------- | :--------------------- | :------------------- |
> > > > | **LLM3 (backtrack)**    | 8                       | 18                          | 74              | 74                     | 22                   |
> > > > | **LLM3 (from scratch)** | 10                      | 10                          | 72              | 72                     | 12                   |
> > > > | **InfoSeeker (ours)**   | **76**                  | **82**                      | **80**          | **82**                 | **40**               |
> > > >
> > > > | Gemini 2.5 Flash          | Robot Navigation-Basic | Robot Navigation-Perturbed | Stack Multiple Block-Basic | Stack Multiple Block-Perturbed | Stack Single Block-Basic | Stack Single Block-Perturbed |
> > > > | :---------------------- | :--------------------- | :------------------------- | :------------------------- | :----------------------------- | :----------------------- | :--------------------------- |
> > > > | **LLM3 (backtrack)**    | **100**                | 6                          | **100**                    | **100**                        | **100**                  | 96                           |
> > > > | **LLM3 (from scratch)** | **100**                | 24                         | **100**                    | **100**                        | **100**                  | **100**                      |
> > > > | **InfoSeeker (ours)**   | **100**                | **100**                    | 98                         | 98                             | **100**                  | **100**                      |

---

### Comment · Area_Chair_A2Rx · 2025-11-24

Dear Reviewer,

Thanks for taking the time to review this work. The authors have responded to your reviews. Can you please have a look at the rebuttal and discuss with the authors?

Best Regards,

AC

---

### Author Response · Authors · 2025-12-03
**Rebuttal Summary**

**Dear Area Chairs and Senior Area Chairs,** we sincerely appreciate your time and effort in reviewing our manuscript. The following is a brief summary of our paper and our rebuttal.

We propose InfoSeeker, an LLM-based decision-making framework that integrates task-oriented planning with information seeking to align internal dynamics and make optimal decisions under uncertainty in observations and environmental dynamics. We also introduce a novel benchmark suite of partially observable environments with incomplete observations and uncertain dynamics. Experiments show that InfoSeeker achieves a **74% absolute performance gain** over prior methods on our benchmark and **outperforms the SOTA baselines** on two well-established public benchmarks, LLM3 and TravelPlanner.

---
**Strengths Recognized by Reviewers:**
1. InfoSeeker addresses uncertainty beyond observations, extending to environmental dynamics, an under-evaluated challenge in prior work. **(gTmW, t7hd, WjYP)**
2. Experiments show promising results, highlighting InfoSeeker's effectiveness across different tasks. **(gTmW, t7hd, WjYP)**
3. The formalization of the problem in terms of the POMDP framework is meaningful. **(t7hd, TXuA)**
---
**Major Concerns and Brief Responses:**
1. Q: What is the efficiency in terms of **token consumption**? **(gTmW, WjYP)**

   A: InfoSeeker **outperforms all baselines** by a substantial margin while using a **similar number of tokens** as the two LLM3 variants, demonstrating both superior effectiveness and efficiency.
2. Q: How does the performance of InfoSeeker vary when **using different LLMs**? **(gTmW)**

   A: Across GPT-4o, o4-mini, gemini-2.0-flash, and gemini-2.5-flash, **InfoSeeker consistently outperformed all baselines**, demonstrating strong robustness to the underlying LLM.
3. Q: What **distinguishes InfoSeeker** from existing LLM-based planning and reasoning methods? **(WjYP, TXuA)**

   A: InfoSeeker **actively performs information-seeking actions to acquire new observations** and align its internal dynamics with the environment before planning. This stands in stark contrast to existing methods, which revise their plans **solely based on past execution traces**.
4. Q: Comparison with LLM reasoning methods. **(WjYP)**

   A: Compared with ReAct+Reflexion, Tree of Thought, and Self-consistency, InfoSeeker not only **achieves the highest success rates** but also **consumes far fewer tokens**, demonstrating both superior performance and efficiency.
5. Q: How can InfoSeeker be applied in **more complex and dynamic real-world environments**? **(TXuA)**

   A: We evaluated InfoSeeker on two challenging and practical benchmarks: LLM3 and TravelPlanner. LLM3 assesses tasks that require controlling a robotic arm in a physical simulator to pack packages, while TravelPlanner evaluates internet-tool use for realistic travel planning scenarios. **InfoSeeker achieved SOTA performance on both practical benchmarks**, highlighting its robustness and adaptability across diverse real-world tasks.

---
**Summary:** We engaged in active discussions with reviewers. More questions and details can be found in the following responses.

**Reviewer gTmW (Initial Rating 4):** The reviewer stated that **"I am happy with the response and it answers all the questions I had. I would like to move my rating up (from 4 to 6)."**

**Reviewer t7hd (Initial Rating 6):** No response yet.

**Reviewer WjYP (Initial Rating 4):** No response yet.

**Reviewer TXuA (Initial Rating 4):** No response yet.

---

### Meta-Review · Area_Chair_uzhv · 2026-01-07

**Summary:**

The submission introduces an LLM-based decision making framework that operates in partially observable environments. The LLMs are tasked to design small diagnostic trials and revise task plans accordingly. The authors also introduced a new text-based benchmark for evaluation, which has uncertainty in both observations and environmental dynamics.

The submission received an initial rating of 6444, with reviewer gTmW commented to upgrade the rating from 4 to 6 after rebutall discussions. Reviewers WjYP and TXuA did not engage in rebuttal, and initially raised concerns about novelty, token consumption, missing baselines, evaluation on real-world environments, and questions on some experimental results. Additionally, reviewer gTmW found the connection to POMDPs interpretive, and the claimed contributions should be adjusted accordingly. Upon checking the authors' rebuttal, the AC believes while many of the raised the concerns have been addressed by the rebuttal, a central concern on the lack of realistic evaluations remains. This concern was also highlighted by reviewer t7hd ("... proposed benchmark is limited and feels hand-crafted ..."). This limitation, along with the other promised revisions to be implemented by the authors, lead the AC to believe that the submission would benefit significantly from a major revision and resubmission, and thus not ready to be accepted by ICLR.


and the submission, assuming all promised revisions will be implemented in the camera ready, passes the acceptance bar of ICLR. The AC therefore recommends acceptance.

**Reviewer Concerns:**

The authors provided extensive additional evaluations that addressed most of the concerns from reviewer gTmW;

Reviewer t7hd raised questions about the types of uncertainty in dynamics considered by the submission, the proposed benchmark being limited and "hand-crafted", and real-world planning scenarios (that requires continuous dynamics), these concerns were acknowledged by the authors as limitations or future work;

Reviewer WjYP had concerns on limited novelty, missing baselines, and some clarification questions. These have been mostly addressed by the rebuttal ("novelty" is often subjective, but the authors clarified that a main distinction is that the proposed framework actively seeks information);

Reviewer TXuA shared a similar concern on the simplicity of the framework, and crucially its performance on more complex and dynamic real-world environments. The authors attempted to mitigate the concern via the diversity (as opposed to the complexity) of the benchmark, which unfortunately doesn't address the concern properly.

**Reviewer Scores:**

The AC believes that reviewer gTmW will change the score to 6, t7hd will keep the score of 6, WjYP may upgrade the score from 4 to 6 or keeps their score (afterall "novelty" can be subjective), and TXuA will keep their score of 4.

---

### Decision · Program_Chairs · 2026-01-26

Reject